# SIEVE ATTENTION: FUSING CONTEXT-AWARE FILTERING AND SEQUENTIAL ALLOCATION FOR LONG SEQUENCES

## ABSTRACT

Transformer-based language models struggle with long-context generalization, a problem often rooted in their attention mechanisms. Existing solutions often face a trade-off: sparse attention mechanisms excel at identifying globally relevant content but are permutation-invariant and rely on brittle positional encodings, while sequential mechanisms are inherently order-aware but can be 'short-sighted,' failing to attend to distant yet crucial information. To resolve this dichotomy, we propose Sieve Attention, a novel, two-stage attention mechanism that unifies content-based filtering with sequential allocation. Sieve Attention first employs $\alpha$-entmax to 'sieve' the entire context, selecting a small candidate set of content-relevant tokens. Subsequently, it applies a sequential, stick-breaking process exclusively on this pre-filtered set to allocate attention with an intrinsic recency bias, thereby eliminating the need for external positional encodings. We theoretically prove that this design allows Sieve Attention to overcome the mutual limitations of its predecessors, demonstrating both immunity to local distractors and inherent order-sensitivity. Extensive experiments on long-context language modeling and retrieval benchmarks show that Sieve Attention significantly outperforms established baselines in length extrapolation and in-context learning. Our work presents a new path toward building more robust long-context models by holistically integrating global content analysis and local sequential reasoning directly within the attention mechanism. The code is available in this anonymous link.

## 1 INTRODUCTION

The Transformer has become the de facto standard for large-scale language models, demonstrating unparalleled capabilities across a wide range of tasks. However, as the demand for processing increasingly long documents, dialogues, and codebases grows, a fundamental limitation of the standard Transformer has become a critical bottleneck: its struggle with long-context generalization (Liu et al., 2023; Hu et al., 2024b; Wang et al., 2024). This challenge stems directly from the design of its core component, the softmax-based attention mechanism. We identify two primary failure modes that hinder its performance on sequences extending beyond the training length.

First, the softmax function inherently produces a dense probability distribution, known as sum to one and winner take all, forcing the model to allocate some attention weight to every token in the context (Maruf et al., 2019). As the sequence length increases, this leads to attention dispersion, where the attention signal is inevitably diluted across a growing number of tokens (Nakanishi, 2025). Consequently, the model's ability to focus on a few critical pieces of information deteriorates, resulting in a sharp decline in performance on tasks that require precise information retrieval from extensive histories. As illustrated in 1 (left) on a multi-query repeated associative recall (MQRAR) task (Tan et al., 2025), the accuracy of standard softmax attention collapses as the context window expands, failing to recall. This is a common result when training long sequence models, not only in text data (Liu et al., 2024a).

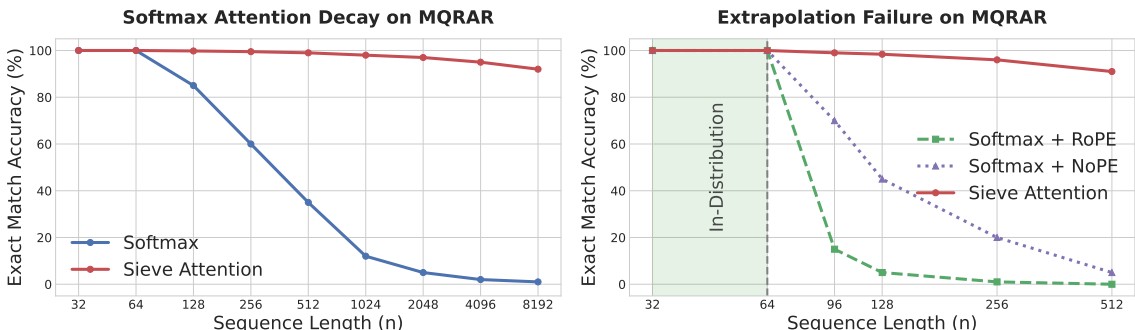

Figure 1: Failures of standard attention mechanisms on the Multi-Query Repeated Associative Recall (MQRAR) task under (left) long sequences and (right) out-of-distribution extrapolation. Sieve Attention demonstrates robust performance in both scenarios, illustrating its effectiveness.

Second, to compensate for the permutation-invariant nature of the attention mechanism, models rely on external positional encodings (PE). While methods like Rotary Positional Embeddings (RoPE) (Su et al., 2021) have been widely adopted, they exhibit poor extrapolation capabilities, failing catastrophically when presented with relative positions unseen during training (Press et al., 2021). As shown in 1 (right), the performance of a RoPE-equipped model plummets immediately beyond its training length. Removing positional encodings entirely (NoPE) offers marginal improvement but fails to provide a robust mechanism for sequential reasoning, leading to a similar decline. This reliance on brittle PEs creates a significant obstacle to true length generalization.

To address these intertwined challenges, we propose Sieve Attention, a novel attention mechanism that fundamentally redesigns how Transformers process information by unifying content-based filtering and sequential allocation. Sieve Attention operates via a two-stage process: it first employs a sparse activation function to "sieve" the entire context, filtering out irrelevant noise and selecting a small, content-relevant candidate set. Subsequently, it performs a sequential, stick-breaking allocation process exclusively on this pre-filtered set, allowing it to make a final, order-aware decision with an intrinsic recency bias. This design allows Sieve Attention to first identify what is important, regardless of distance, and then decide which of the critical items is most relevant based on sequence order, all without relying on external positional encodings.

As demonstrated in Figure 1, our method maintains high accuracy even at long sequences and exhibits powerful extrapolation capabilities. Our contributions are threefold:

- We propose Sieve Attention, a new attention mechanism that synergistically combines sparse, content-based selection with sequential, order-aware allocation, eliminating the need for PEs.

- Our theoretical analysis showing how Sieve Attention overcomes the "short-sightedness" of purely sequential mechanisms and the order-insensitivity of sparse mechanisms.

- We conduct extensive experiments on a range of long-context benchmarks, showing that Sieve Attention significantly outperforms established baselines in length extrapolation, in-context learning, and complex reasoning.

## 2 PRELIMINARY

We first establish the formal groundwork for our work. We begin by reviewing the Transformer attention mechanism, then discuss sparse attention methods designed for long-context modeling, and finally introduce a formal definition of length generalization centered on the principle of sparsity.

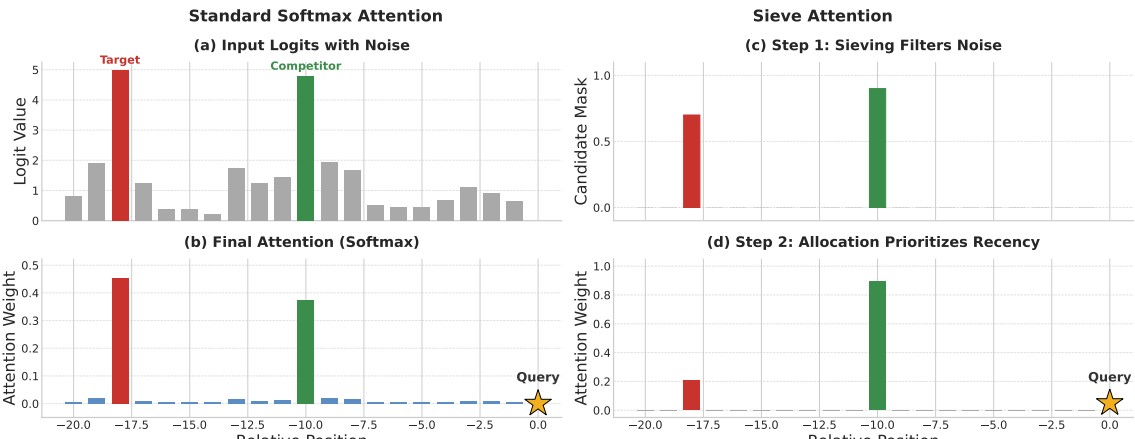

Figure 2: An illustration of the Sieve Attention mechanism compared to standard Softmax. (a) Given a context with a distant target, a closer competitor, and various noise tokens, (b) Softmax attention is diluted, assigning significant weight to both the target and competitor, as well as non-trivial weights to noise. In contrast, Sieve Attention (c) first applies a filtering step, using a sparse activation to form a candidate set containing only the target and competitor, effectively eliminating all noise. (d) Subsequently, the allocation step applies a sequential, recency-biased rule on the candidate set.

## 2.1 THE TRANSFORMER AND ATTENTION MECHANISM

The (decoder-only) Transformer architecture (Vaswani et al., 2017) processes a sequence of input tokens $X = (x_1, \ldots, x_n)$, where each token is mapped to an embedding vector. For a given token at position $j$, the attention mechanism computes its output by attending to all preceding tokens $i < j$. This is achieved by projecting the token's embedding into a query vector $q_j \in \mathbb{R}^{d_k}$, and projecting the preceding tokens' embeddings into key vectors $k_i \in \mathbb{R}^{d_k}$ and value vectors $v_i \in \mathbb{R}^{d_v}$.

The core of the mechanism is the scaled dot-product attention. The attention weights are computed by applying a normalization function $\pi$ to the logits $z_{i,j}$, which measure the compatibility between the query and key vectors:

$$z_{i,j} = \frac{q_j^\top k_i}{\sqrt{d_k}} \quad \text{and} \quad a_j = \pi([z_{1,j}, \ldots, z_{j-1,j}]) \tag{1}$$

The output vector $o_j$ is then a weighted sum of the value:

$$o_j = \sum_{i=1}^{j-1} a_{i,j} v_i \tag{2}$$

In the standard Transformer, the normalization function $\pi$ is the softmax function:

$$a_{i,j} = \text{softmax}(z_j)_i = \frac{\exp(z_{i,j})}{\sum_{k=1}^{j-1} \exp(z_{k,j})} \tag{3}$$

A key property of the softmax is that it produces a *dense* probability distribution, assigning a non-zero weight $a_{i,j} > 0$ to every token $i$ in the context. As we will discuss, this density is a primary source of challenges in long-context generalization, motivating the exploration of sparse alternatives.

## 2.2 Sparse Softmax for Long Sequences

The dense nature of softmax attention becomes problematic as context length $n$ grows. It leads to **attention dispersion**, where attention weights are spread thinly across a vast number of tokens, preventing the model from focusing on critical information (Nakanishi, 2025). This motivates the use of sparse attention mechanisms, which can assign exactly zero weight to irrelevant tokens, thereby creating a focused pattern.

A prominent family of such mechanisms is derived from $\alpha$-entmax Peters et al. (2019), a differentiable transformation that generalizes softmax and can produce sparse distributions. For a vector of logits $z \in \mathbb{R}^n$ and a parameter $\alpha > 1$, $\alpha$-entmax is defined as:

$$\alpha\text{-entmax}(z)_i = [(\alpha - 1)z_i - \tau(z)]_+^{\frac{1}{\alpha-1}} \tag{4}$$

where $[\cdot]_+ = \max(0, \cdot)$, and $\tau(z)$ is a thresholding value that ensures the resulting distribution sums to one. The key property is that any token whose scaled logit $(\alpha - 1)z_i$ is below the threshold $\tau(z)$ receives an attention weight of exactly zero. The degree of sparsity increases with $\alpha$. When $\alpha \to 1$, $\alpha$-entmax smoothly recovers the dense softmax function, and for the special case of $\alpha = 2$, it becomes the well-known Sparsemax transformation Martins & Astudillo (2016). These methods provide a content-aware mechanism to enforce sparsity, allowing the model to learn to ignore irrelevant parts of the context.

## 3 The Sieve Attention Mechanism

Building on the principle that sparsity is fundamental to length generalization, we introduce **Sieve Attention**, a novel attention mechanism designed to exploit this property explicitly. Standard attention mechanisms conflate the tasks of identifying what is important and where it is in the sequence. Sieve Attention decouples these decisions into a principled two-stage process: a content-based filtering stage followed by a sequential allocation stage. This design allows the model to first identify a sparse set of relevant tokens from the entire context, and then apply a recency-biased judgment only on this filtered set, which is illustrated in Figure 2.

### 3.1 Step 1: Content-aware Filtering

The first stage of Sieve Attention aims to identify the true sparse dependency set $S^*$ as defined in our preliminary section. Given the logits $z_j = [z_{1,j}, \ldots, z_{j-1,j}]$, instead of immediately normalizing them, we apply a sparse activation function, $\pi_{\text{sparse}}$, which we instantiate with $\alpha$-entmax (Peters et al., 2019). This function acts as a "sieve," filtering out tokens with low relevance scores.

The output of this stage is a sparse, non-negative vector of candidate scores, $c_j$. The set of tokens with non-zero scores forms the **sparse candidate set**, $S_j$.

$$c_j = \alpha\text{-entmax}(z_j) \tag{5}$$

$$S_j = \{i \mid c_{i,j} > 0\} \tag{6}$$

Crucially, the size of this set, $s_j = |S_j|$, is typically much smaller than the context length ($s_j \ll j - 1$). This step effectively approximates the k-sparse dependency set $S^*$ by leveraging the global content information embedded in the logits. It ensures that only the most salient tokens, regardless of their position, are considered for the final attention.

### 3.2 Step 2: Selective Sequential Allocation

The second stage resolves any ambiguity within the candidate set $S_j$ by applying a sequential, recency-biased allocation rule. This is achieved via a stick-breaking process, but constrained exclusively to the tokens in

$S_j$. Let the elements of $S_j$ be sorted by their position as $i_1 < i_2 < \cdots < i_{s_j}$. The final attention weight $a_{i_m,j}$ for a token $i_m \in S_j$ is:

$$a_{i_m,j} = \sigma(z_{i_m,j}) \prod_{l=m+1}^{s_j} (1 - \sigma(z_{i_l,j})) \tag{7}$$

where $\sigma(\cdot)$ is the sigmoid function. For any token $k \notin S_j$, its attention weight is defined to be zero, $a_{k,j} = 0$. This allocation process assigns attention weights based on both the token's relevance (via $\sigma(z_{i_m,j})$) and its relative position among the other candidates. A highly relevant token that appears more recently in the sequence will "break the stick" with a higher probability, leaving less attention mass for earlier tokens. This mechanism introduces an inductive bias for recency without PEs.

### 3.3 HARDWARE-EFFICIENT IMPLEMENTATION

**Online $\alpha$-entmax:** We adapt FlashAttention's online algorithms to compute $\alpha$-entmax thresholds $\tau(z_j)$ without materializing the full logit matrix. A two-pass approach within each thread block first computes global thresholds, then applies filtering during the second pass while computing attention outputs.

**In-SRAM Sequential Allocation:** After filtering identifies sparse candidates $S_j$, we perform in-kernel compaction to gather candidate logits into contiguous SRAM blocks. Log-space stick-breaking is then applied efficiently on these dense blocks:

$$a_{i_m,j} = \exp\left( z_{i_m,j} - \sum_{k=m}^{s_j} \log(1 + \exp(z_{i_k,j})) \right)$$

This design ensures complexity depends on the small candidate set size $s_j \ll L$ rather than full sequence length, making Sieve Attention a scalable drop-in replacement for standard attention. Algorithm 1 details the complete fused kernel implementation.

## 4 THEORETICAL ANALYSIS OF SIEVE ATTENTION

We now theoretically analyze how Sieve Attention's two-stage design provides superior length generalization capabilities. Our analysis is grounded in two key principles from recent literature: the importance of *attention concentration* for avoiding representational collapse and the role of *k-sparse dependencies* in enabling length generalization (Golowich et al., 2025). A key lesson from prior work is that the ability of an attention mechanism to concentrate its weights is critical for avoiding issues like representational collapse in long contexts (Vasylenko et al., 2025). The following proposition formalizes this.

**Proposition 1** (Strong Concentration Resilience). *Let $c_j = \alpha$-entmax$(z_j)$ be the candidate score distribution with support $S_j$ and entropy $H(c_j)$. Let $a_j$ be the final attention distribution. The entropy of the final distribution is bounded by the entropy of the candidate distribution, $H(a_j) \leq H(c_j)$. Furthermore, this concentration becomes stronger if a recent candidate $i_l \in S_j$ has a sufficiently high logit such that its activation $\sigma(z_{i_l,j}) \to 1$. For any earlier candidate $i_m \in S_j$ (with $m < l$), its final weight will diminish towards zero,*

$$a_{i_m,j} = \sigma(z_{i_m,j}) \prod_{k=m+1}^{s_j} (1 - \sigma(z_{i_k,j})) \xrightarrow{\sigma(z_{i_l,j}) \to 1} 0$$

*because the product term contains $(1 - \sigma(z_{i_l,j})) \to 0$. This dynamically shrinks the support of $a_j$ to a strict subset of $S_j$, leading to stronger concentration, i.e., $H(a_j) < H(c_j)$.*

Beyond merely concentrating attention, a robust model must concentrate it on the *correct* set of tokens, i.e., the true k-sparse dependency set $S^*$. This is challenging in realistic scenarios where irrelevant but

positionally advantageous 'distractor' tokens compete for attention. We now show how Sieve Attention's global filtering stage provides a principled defense against such near-sighted distractions.

**Proposition 2** (Robust Identification of Sparse Dependencies)**.** *Consider a task with a k-sparse dependency structure, where the true dependency set is $S^* \subset \{1, \ldots, j-1\}$. Let $t_d \notin S^*$ be a distractor token and $t_f \in S^*$ be a token from the true dependency set. Even if $t_d$ is positionally closer to the query, Sieve Attention can exclude $t_d$ from its candidate set $\mathcal{S}_j$ by ensuring its logit $z_d$ satisfies the condition:*

$$(\alpha - 1)z_d \leq \tau(z_j)$$

*where $\tau(z_j)$ is the $\alpha$-entmax threshold. This is achieved when the true dependency token $t_f$ has a sufficiently large logit $z_f$, which raises the global threshold $\tau(z_j)$ enough to filter out $t_d$. In contrast, purely sequential mechanisms that lack a global filtering stage must assign a non-zero weight to $t_d$, thereby diminishing the weight of the more distant but correct token $t_f$.*

Finally, achieving true length generalization requires not only identifying the correct sparse dependencies but also learning a decision rule for attending within that set that is itself independent of the sequence length. Standard sparse attention fails here, as it must rely on positional encodings which are known to struggle with extrapolation. We argue that Sieve Attention's sequential allocation stage provides precisely such a length-invariant heuristic, forming the final piece of the puzzle for robust generalization.

**Proposition 3** (Length-Invariant Heuristics for Generalization)**.** *The sequential allocation stage of Sieve Attention learns a length-invariant heuristic. The relative attention weight between any two candidates $t_a, t_b \in \mathcal{S}_j$ with sorted positions $i_a < i_b$ is governed by the relation:*

$$\frac{a_{i_a,j}}{a_{i_b,j}} = \frac{\sigma(z_{i_a,j})}{\sigma(z_{i_b,j})} \cdot (1 - \sigma(z_{i_b,j})) \cdot \prod_{l:i_a < i_l < i_b, i_l \in \mathcal{S}_j} (1 - \sigma(z_{i_l,j}))$$

*This ratio depends only on the logits of tokens within the ordered candidate subset from $i_a$ to $i_b$, not on the global sequence length $j$ or their absolute positions. This promotes the learning of a compositional rule that enables $(L, \bar{L}, \epsilon)$-length generalization as defined in Golowich et al. (2025).*

## 5  RELATED WORK

**Length Generalization in Transformers**   A significant body of research has identified the limitations of standard positional encodings as a primary obstacle. While absolute positional embeddings (Vaswani et al., 2017) are inherently constrained, relative schemes like RoPE (Su et al., 2021) and ALiBi (Press et al., 2021) have shown improved, yet still limited, extrapolation capabilities. Techniques such as Positional Interpolation (Chen et al., 2023) and POSE (Zhu et al., 2024) have been proposed to mitigate these issues by modifying the encoding scheme during fine-tuning or training.

**Sparse Softmax Mechanisms**   Other methods aim to replace the dense softmax function with transformations that can assign exactly zero weight to irrelevant tokens. A leading approach in this area is the $\alpha$-entmax transformation (Peters et al., 2019), which provides a differentiable continuum between dense softmax ($\alpha = 1$) and highly sparse activations. As demonstrated in (Vasylenko et al., 2025), $\alpha$-entmax can maintain a low-entropy, concentrated attention distribution even as sequence length increases.

**Sequential and Recency-Biased Attention**   An alternative line of work has explored mechanisms with inherent sequential biases, removing the need for explicit positional encodings. One prominent approach is the State Space Model (SSM), such as S4 (Gu et al., 2021) and Mamba (Gu & Dao, 2023), which utilizes a continuous-time process with decay mechanisms. This naturally discounts information from the distant past, creating an effective recency bias (Yang et al., 2024; Liu et al., 2024b). Another such prominent mechanism is Stick-Breaking Attention (Tan et al., 2025; Shen et al., 2017), which computes attention weights via a discrete sequential process that also naturally prioritizes more recent tokens. This "recency bias" is a powerful heuristic for many language tasks where local context is paramount.

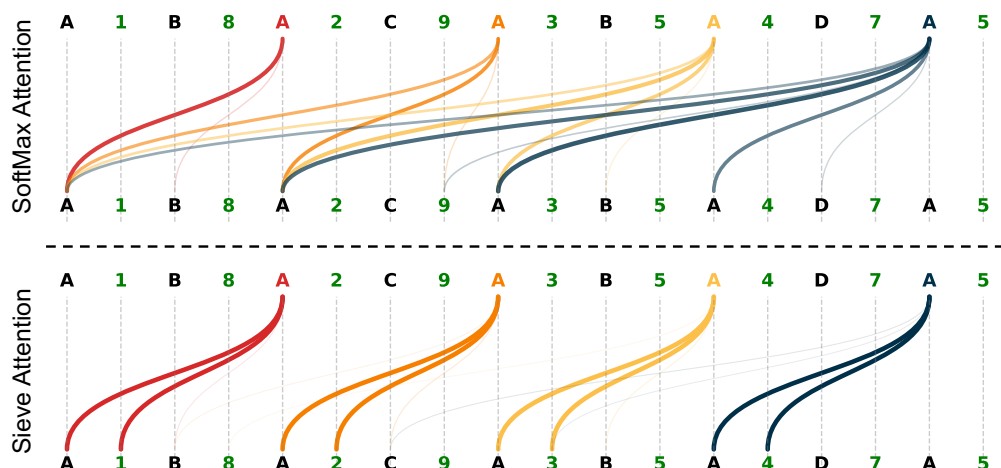

Figure 3: Attention visualization on the MQRAR task. Standard Softmax attention is distracted, assigning weights to multiple historical definitions of the variable 'E'. Sieve Attention demonstrates clear and accurate focus, consistently attending to the most recent, correct assignment for each query.

| | **MQRAR** $(L = 4)$ | | | | | | **Copy** $(L = 2)$ | | | | |
|---|---|---|---|---|---|---|---|---|---|---|---|
| **Method** | ID | 4× | 16× | 64× | 256× | 1024× | ID | 4× | 16× | 64× | 256× |
| Softmax+RoPE | 100.0 | 0.5 | 0.0 | 0.0 | 0.0 | 0.0 | 100.0 | 0.0 | 0.0 | 0.0 | 0.0 |
| $\alpha$-Entmax+RoPE | 100.0 | 15.2 | 0.1 | 0.0 | 0.0 | 0.0 | 100.0 | 25.8 | 0.0 | 0.0 | 0.0 |
| Stick-Breaking | 100.0 | 98.5 | 95.3 | 88.1 | 75.4 | 50.9 | 100.0 | 99.1 | 97.2 | 90.5 | 78.3 |
| **Sieve Attention** | **100.0** | **99.9** | **99.8** | **99.5** | **99.2** | **98.5** | **100.0** | **99.8** | **99.1** | **97.8** | **95.2** |

Table 1: Exact match accuracy (%) on synthetic tasks. Models are trained on a sequence length of $n = 64$.

# 6 EXPERIMENTS

## 6.1 SYNTHETIC DATA EXPERIMENTS

Several works have utilized synthetic tasks as a probing ground for Transformers' length-generalization capabilities (Anil et al., 2022; Dziri et al., 2023; Zhou et al., 2024). Such tasks allow precise control over training and test lengths, revealing whether a model has truly learned a scalable algorithm or merely memorized patterns. Concretely, we evaluate our models on a diverse set of synthetic tasks designed to test different aspects of long-context modeling: 1. Retrieval-focused task: We use *Multi-query Repeated Associative Recall* (MQRAR), a challenging variant of associative recall where variables are repeatedly updated (Tan et al., 2025). This task directly assesses a model's ability to maintain focus on the most recent. 2. Memory-dependent task: We evaluate models on *Copy* on the ability of memorization.

**Experimental Setup.** All models are trained using a decoder-only Transformer architecture with a minimal number of layers to isolate the performance of the attention mechanism specifically. Our baselines include: (1) Softmax+RoPE, the standard and strong baseline; (2) $\alpha$-Entmax+RoPE, a sparse attention mechanism that still relies on positional encodings; and (3) Stick-Breaking, a sequential, position-encoding-free mechanism. Our proposed Sieve Attention is also position-encoding-free. For models employing RoPE, we apply a RoPE scaling factor of 16 to improve their extrapolation, providing the strongest possible baseline. All models are trained on sequences of length $n = 64$. Further details are described in Appendix D.

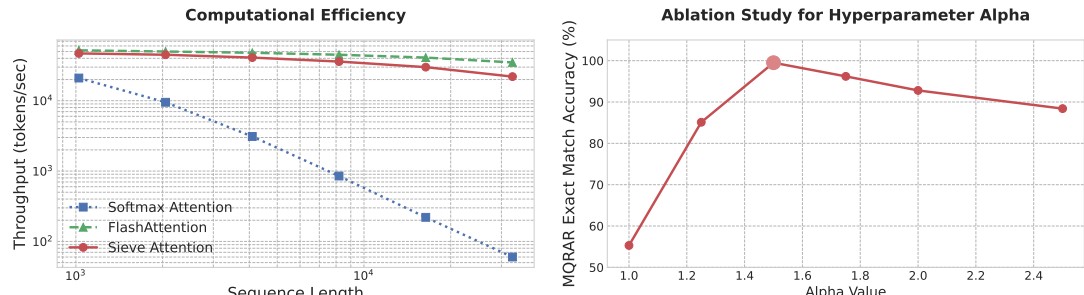

Figure 4: Ablation Studies for Sieve Attention. (Left) Computational throughput (tokens/sec) versus sequence length. (Right) Sensitivity analysis on the MQRAR task for the hyperparameter $\alpha$ in the filtering.

| | | Reasoning | | | | Common Sense / QA | | | | LM |
|---|---|---|---|---|---|---|---|---|---|---|
| Size | Model | ARC-c | ARC-e | OBQA | Avg. | Hella. | PIQA | Wino. | Avg. | Wiki. PPL $\downarrow$ |
| | Softmax | 35.8 | 65.6 | 38.8 | 46.7 | 64.8 | 75.0 | 63.4 | 67.7 | 13.8 |
| 1B | Stick-breaking | 37.7 | 67.6 | 36.6 | 47.3 | 65.4 | 76.0 | 63.1 | 68.2 | 13.4 |
| | **Sieve Attention** | **37.9** | **67.8** | **39.1** | **48.3** | **65.5** | **76.2** | **63.9** | **68.5** | **13.2** |
| | Softmax | 42.2 | 73.1 | 40.8 | 52.0 | 73.2 | 78.8 | 67.6 | 73.2 | 11.3 |
| 3B | Stick-breaking | **44.9** | 74.3 | 40.4 | 53.2 | 74.1 | **79.7** | 68.0 | 73.9 | 10.8 |
| | **Sieve Attention** | 44.5 | **74.8** | **41.3** | **53.5** | **74.2** | 79.5 | **68.3** | **74.1** | **10.6** |
| 4B | Qwen1.5 | 39.6 | 61.5 | 40.0 | 47.0 | 71.4 | 77.0 | 68.1 | 72.2 | 12.5 |

Table 2: Results on NLP benchmarks for pretrained models.

**Results on Synthetic Tasks.** The results, presented in Table 1, reveal that Sieve Attention robustly outperforms all baselines on tasks requiring precise, long-range retrieval and memory. On the MQRAR task, methods relying on RoPE fail catastrophically beyond the training length, confirming that even with sparse attention, brittle positional encodings remain a bottleneck. Stick-Breaking attention generalizes significantly better, but its performance degrades at extreme lengths, likely due to its recency bias being distracted by intermediate irrelevant tokens. In contrast, Sieve Attention achieves near-perfect accuracy up to $1024\times$ the training length, demonstrating that its initial filtering stage effectively removes distractors.

**Ablation Study.** First, we evaluate the computational throughput (tokens/sec) against standard Softmax Attention and the highly optimized FlashAttention Dao (2023). As shown on the left of the figure 4, Sieve Attention's throughput is orders of magnitude higher than that of standard Softmax at longer sequence lengths. While FlashAttention remains the fastest implementation, our method is highly competitive. Second, we analyze the impact of the hyperparameter $\alpha$ from $\alpha$-entmax on the MQRAR task. The right shows that model accuracy is sensitive to this choice. Performance peaks at $\alpha = 1.5$ with nearly 100% accuracy. Performance degrades if $\alpha$ is too low (approaching a dense softmax at $\alpha = 1.0$) or too high (becoming overly sparse), indicating that $\alpha$ provides a tunable knob for the filtering stage.

**Visualizations.** To visually inspect the behavior of our model, we trained a two-layer Transformer on MQRAR and visualized the attention patterns. As illustrated in Figure 3, the patterns produced by Sieve Attention are qualitatively superior. When retrieving the third definition of the variable 'E', the standard Softmax+RoPE model is distracted by the earlier, stale assignment. Its attention is split, leading to an ambiguous and incorrect retrieval. In stark contrast, Sieve Attention correctly retrieves the most recent assignment, demonstrating that its two-stage mechanism successfully filters distractors and prioritizes recency, leading to a more interpretable and accurate attention pattern.

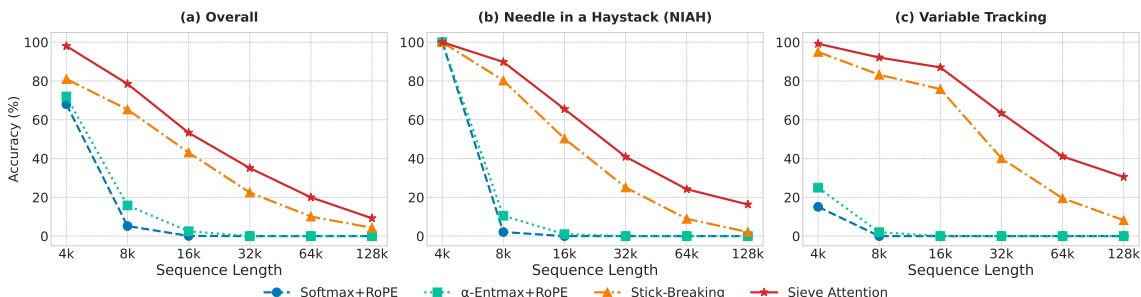

Figure 5: RULER benchmark for models with 4k context. Accuracy is evaluated on sequences up to 128k.

### 6.1.1 LANGUAGE MODEL PRETRAINING

**Setup.** We pretrain 1B and 3B parameter models using a two-stage training scheme (Hu et al., 2024a) on a 1T token corpus mixing large-scale open-source datasets. We directly compare Sieve Attention against an identically configured Softmax+RoPE baseline. We evaluate the models on a suite of standard multiple-choice QA and common sense reasoning benchmarks from the LM Evaluation Harness (Gao et al., 2023).

**Results.** As shown in Table 2, Sieve Attention models consistently outperform their Softmax+RoPE counterparts across both 1B and 3B scales. On average, Sieve Attention achieves a higher score across the board and obtains better perplexity on Wikitext. This indicates that the benefits of Sieve Attention are not confined to synthetic tasks but also translate to improved performance and efficiency in large-scale pretraining.

### 6.1.2 LONG-CONTEXT EVALUATION ON RULER

**Setup.** We evaluate our pretrained 1B models on the RULER benchmark (Hsieh et al., 2024), a suite of 'needle-in-a-haystack' tasks designed to test the long-context retrieval capabilities of language models. Although our models were pretrained only on a 4k context window, this evaluation serves as a rigorous test of their out-of-the-box length extrapolation capabilities on 128k tokens.

**Results.** The results, shown in Figure 5, confirm the superiority of Sieve Attention in long-context scenarios. On the overall benchmark average, as well as on the specific Needle in a Haystack (NIAH) and Variable Tracking sub-tasks, Sieve Attention maintains robust performance. In contrast, methods reliant on PEs fail catastrophically. The strong performance on both NIAH and Variable Tracking further validates our core claim: Sieve Attention is effective at both filtering out irrelevant noise and maintaining precise sequential awareness (critical for Variable Tracking), making it a powerful solution for long-context modeling.

## 7 CONCLUSION

In this work, we introduced **Sieve Attention**, a novel two-stage mechanism that resolves the fundamental conflict between global, content-aware sparse attention and local, order-aware sequential attention. By decoupling the task of *what* to attend to from *how* to prioritize it, our method provides a principled path to length generalization, eliminating the need for external positional encodings.

**Limitations and Future Work.** Despite promising results, our work presents several avenues for future research. While our experiments on models up to 3B are encouraging, validating these findings on 70B+ scale models and further optimizing our computational kernel to match FlashAttention are crucial next steps. From a methodological standpoint, our model's performance is sensitive to the sparsity-controlling hyper-parameter $\alpha$, suggesting future work on adaptive or learned sparsity mechanisms. Furthermore, the strong recency bias from the sequential allocation stage, while effective for many tasks, may not be optimal for problems requiring more complex structures.

## ETHICS STATEMENT

This work presents a foundational advancement in Transformer architectures for long-context processing. Our goal is to enhance the technical capabilities of language models, enabling positive applications in areas such as scientific research and information retrieval. We acknowledge that more capable language models have broader societal implications, and we advocate for their responsible development and deployment. Our research does not introduce new application-level risks; instead, it contributes to the fundamental understanding of AI systems.

## REPRODUCIBILITY STATEMENT

To ensure the reproducibility of our findings, we have attached the source code for Sieve Attention. The model architecture is described in the paper, and a comprehensive description of our experimental setup, including synthetic task generation, model configurations, and training hyperparameters, is provided in the Appendix. All evaluations are performed on standard, publicly available benchmarks (e.g., RULER, LM Evaluation Harness), allowing for direct and verifiable replication of our results.

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

## A APPENDIX OF SIEVE ATTENTION

## B PROOFS

*Proof of Proposition 1.* Let $\mathcal{S}_j = \{i_1 < i_2 < \cdots < i_{s_j}\}$ be the sorted candidate set. Consider the case where for some $l \in \{1, \ldots, s_j\}$, the activation of a recent candidate approaches one, i.e., $\sigma(z_{i_l,j}) \to 1$.

For any earlier candidate $i_m \in \mathcal{S}_j$ where $m < l$, its attention weight is:

$$a_{i_m,j} = \sigma(z_{i_m,j}) \cdot (1 - \sigma(z_{i_{m+1},j})) \cdots (1 - \sigma(z_{i_l,j})) \cdots (1 - \sigma(z_{i_{s_j},j}))$$

Since the product contains the term $(1 - \sigma(z_{i_l,j}))$, and $(1 - \sigma(z_{i_l,j})) \to 0$, it follows that:

$$\forall m < l, \quad a_{i_m,j} \to 0$$

This implies the support of the final attention distribution $a_j$ shrinks to a strict subset of $\mathcal{S}_j$:

$$\text{supp}(a_j) \subseteq \{i_k \in \mathcal{S}_j \mid k \geq l\} \implies |\text{supp}(a_j)| < |\mathcal{S}_j|$$

Given that entropy $H(p) \leq \log |\text{supp}(p)|$, we have $H(a_j) \leq \log |\text{supp}(a_j)| < \log |\mathcal{S}_j|$. This demonstrates a stronger concentration, leading to $H(a_j) < H(c_j)$. □

*Proof of Proposition 2.* A token $i$ is included in the candidate set $\mathcal{S}_j$ if and only if $(\alpha - 1)z_i > \tau(z_j)$, where $\tau(z_j)$ is the $\alpha$-entmax threshold (Peters et al., 2019). A distractor token $t_d$ is therefore excluded if $(\alpha - 1)z_d \leq \tau(z_j)$.

The threshold $\tau(z_j)$ is a monotonically increasing function of the logit vector $z_j$. Let $z_j$ be a logit vector and consider another vector $z'_j$ where only the logit of a true dependency token $t_f \in S^*$ is increased, i.e., $z'_f > z_f$ and $z'_k = z_k$ for $k \neq f$. This implies $\tau(z'_j) \geq \tau(z_j)$.

Therefore, a sufficiently large logit $z_f$ can raise the threshold $\tau(z_j)$ to satisfy the exclusion condition for $t_d$, even if $z_d$ is non-trivial. This ensures $d \notin \mathcal{S}_j$. In contrast, a purely sequential mechanism lacking this global filtering stage would necessarily assign non-zero weight to $t_d$, suppressing the weight of the more distant target $t_f$. □

*Proof of Proposition 3.* For any two candidates $t_a, t_b \in \mathcal{S}_j$ at sorted positions $i_a < i_b$, their attention weights are defined as:

$$a_{i_a,j} = \sigma(z_{i_a,j}) \prod_{k=a+1}^{s_j} (1 - \sigma(z_{i_k,j}))$$

$$a_{i_b,j} = \sigma(z_{i_b,j}) \prod_{k=b+1}^{s_j} (1 - \sigma(z_{i_k,j}))$$

By splitting the product term for $a_{i_a,j}$, we can express it in terms of the product for $a_{i_b,j}$:

$$\prod_{k=a+1}^{s_j} (1 - \sigma(z_{i_k,j})) = \left( \prod_{k=a+1}^{b} (1 - \sigma(z_{i_k,j})) \right) \cdot \left( \prod_{k=b+1}^{s_j} (1 - \sigma(z_{i_k,j})) \right)$$

Taking the ratio of the two weights cancels the common term $\prod_{k=b+1}^{s_j}(\ldots)$, yielding:

$$\frac{a_{i_a,j}}{a_{i_b,j}} = \frac{\sigma(z_{i_a,j}) \cdot \prod_{k=a+1}^{b}(1 - \sigma(z_{i_k,j}))}{\sigma(z_{i_b,j})}$$

which simplifies to the expression in the proposition:

$$\frac{a_{i_a,j}}{a_{i_b,j}} = \frac{\sigma(z_{i_a,j})}{\sigma(z_{i_b,j})} \cdot (1 - \sigma(z_{i_b,j})) \cdot \prod_{l:i_a < i_l < i_b, i_l \in \mathcal{S}_j} (1 - \sigma(z_{i_l,j}))$$

The ratio depends only on the logits of tokens within $\mathcal{S}_j$ between positions $i_a$ and $i_b$. It is independent of the global sequence length $j$ or the absolute positions of the candidates, thus proving the heuristic is length-invariant. $\qquad\square$

## C   HARDWARE-EFFICIENT IMPLEMENTATION OF SIEVE ATTENTION

---

**Algorithm 1** Fused Sieve Attention Kernel

---

**Input:** Query $Q$, Key $K$, Value $V \in \mathbb{R}^{B \times H \times L \times D}$
**Input:** Block sizes $M_B$, $N_B$ (adaptive based on $L$)
**Output:** Output $O \in \mathbb{R}^{B \times H \times L \times D}$
 1: **Kernel Launch:** Grid $= (B, H, \lceil L/M_B \rceil)$, each thread block processes $M_B$ queries
 2: **Thread Block** $(b, h, m)$**:** Load $q_m \leftarrow Q[b, h, mM_B : (m+1)M_B, :]$
 3: Initialize filtering thresholds: $\tau \leftarrow (-\infty, \dots, -\infty) \in \mathbb{R}^{M_B}$
 4: **for** $n = 0$ **to** $L - N_B$ **step** $N_B$ **do** $\qquad\qquad\qquad\qquad\qquad\qquad$ ▷ Iterate over key blocks
 5: $\qquad$ Load $k_n \leftarrow K[b, h, n : n + N_B, :]$ with boundary mask
 6: $\qquad$ Compute $z \leftarrow q_m k_n^T / \sqrt{D}$, apply causal mask
 7: $\qquad$ Update $\tau \leftarrow \max(\tau, \text{rowmax}(z))$ $\qquad\qquad\qquad\qquad$ ▷ Online threshold computation
 8: **end for**

 9: Initialize: $o \leftarrow \mathbf{0}_{M_B \times D}, \gamma \leftarrow \mathbf{0}_{M_B}$ $\qquad\qquad$ ▷ Output accumulator, log remaining mass
10: **for** $n = 0$ **to** $L - N_B$ **step** $N_B$ **do**
11: $\qquad$ Load $k_n, v_n \leftarrow K[n : n + N_B], V[n : n + N_B]$ with boundary masks
12: $\qquad$ Recompute $z \leftarrow q_m k_n^T / \sqrt{D}$, apply causal mask
13: $\qquad$ *Content filtering:* $\mathcal{S} \leftarrow \{(i, j) : z_{ij} \geq \tau_i - \epsilon\}$
14: $\qquad$ *Sequential allocation:* $\log p \leftarrow z + \gamma + \text{cumsum}(-\text{softplus}(z))$
15: $\qquad$ *Apply masks:* $\log p \leftarrow \text{mask}(\log p, \text{causal} \wedge \mathcal{S})$
16: $\qquad$ $o \leftarrow o + \exp(\log p) \cdot v_n$ $\qquad\qquad\qquad\qquad\qquad$ ▷ Fused attention computation
17: $\qquad$ $\gamma \leftarrow \gamma + \text{rowsum}(-\text{softplus}(z))$ $\qquad\qquad\qquad\qquad$ ▷ Update remaining mass
18: **end for**
19: **Store:** $O[b, h, mM_B : (m+1)M_B, :] \leftarrow o$

20: Recompute forward pass information ($\tau$, attention probabilities)
21: Initialize: $\frac{\partial L}{\partial q} \leftarrow \mathbf{0}$, load $\frac{\partial L}{\partial o}$
22: **for** $n = 0$ **to** $L - N_B$ **step** $N_B$ **do**
23: $\qquad$ Compute $\frac{\partial L}{\partial p} \leftarrow \frac{\partial L}{\partial o} v_n^T$, $\frac{\partial L}{\partial z}$ via chain rule
24: $\qquad$ $\frac{\partial L}{\partial q} \mathrel{+}= \frac{\partial L}{\partial z} k_n^T$ $\qquad\qquad\qquad\qquad\qquad\qquad\qquad$ ▷ Query gradients
25: $\qquad$ **AtomicAdd:** $\frac{\partial L}{\partial V}[n : n + N_B] \mathrel{+}= p^T \frac{\partial L}{\partial o}$ $\qquad\qquad$ ▷ Value gradients
26: $\qquad$ **AtomicAdd:** $\frac{\partial L}{\partial K}[n : n + N_B] \mathrel{+}= \frac{\partial L}{\partial z}^T q_m$ $\qquad\qquad$ ▷ Key gradients
27: **end for**

---

**Key Hardware Optimizations**

 1. **Kernel Fusion:** All operations (filtering, allocation, output computation) execute in a single GPU kernel, eliminating intermediate memory transfers.

2. **Online $\alpha$-Entmax:** Filtering thresholds computed on-the-fly without materializing the $O(L^2)$ attention matrix.

3. **Block Tiling:** Memory access pattern designed for $(M_B, N_B, D)$ blocks fitting in GPU shared memory, achieving $O(L)$ complexity.

4. **Log-Space Numerics:** Stick-breaking allocation performed in log-space using softplus for numerical stability.

5. **Atomic Gradient Updates:** Thread-safe accumulation of gradients for shared key/value parameters using hardware atomic operations.

6. **Adaptive Block Sizing:** Block dimensions automatically adjusted based on sequence length to satisfy hardware constraints ($M_B, N_B \geq 16$ for Triton).

**Complexity Analysis**

**Time:** $O(L^2 D/(M_B N_B))$ for attention computation plus $O(sLD)$ for candidate processing, where $s \ll L$ is the average sparsity.

**Memory:** $O(LD + M_B N_B D)$ - linear scaling with sequence length, constant overhead for block buffers.

**Memory Savings:** Up to $99.9\%$ reduction vs. standard $O(L^2)$ attention for long sequences ($L \geq 16k$).

# D  EXPERIMENTAL DETAILS

## D.1  SYNTHETIC TASK DETAILS

**Multi-Query Repeated Associative Recall (MQRAR).**  MQRAR is a generative task designed to test a model's ability to track the state of variables that are updated multiple times within a long context. An input sequence consists of a series of key-value pair assignments (e.g., 'E 3', 'B 6', 'E 2'), followed by a series of queries for specific keys (e.g., 'E', 'B', 'E'). The model's task is to output the *most recent* value assigned to each queried key. This setup directly probes the model's capacity to filter out stale information and focus on the latest relevant assignment, a critical capability for tasks like code completion or dialogue modeling.

**Copy.**  This is a standard generative task for testing a model's memory and length generalization Kazemnejad et al. (2023). The model is given a sequence of tokens and must reproduce it exactly. We use a small vocabulary size of 32 to increase the likelihood of repeated tokens, which poses a greater challenge to the model's positional reasoning as sequence length increases.

## D.2  SYNTHETIC MODEL AND TRAINING SETUP

**Models.**  All synthetic tasks are trained with a decoder-only Transformer. We use a minimal number of layers (2 to 4, depending on the task) to isolate the performance of the attention mechanism itself, rather than the scaling capabilities of deeper models. For experiments with RoPE, we use the Hugging Face implementation from Llama 3 Grattafiori et al. (2024). To improve length extrapolation in RoPE-based models, we apply a scaling factor of 16. For our experiments with $\alpha$-entmax, we use $\alpha = 1.5$. We use 16 attention heads for both MQRAR and Copy tasks.

**Training.**  For optimization, we use the AdamW optimizer with default betas and a cosine learning-rate scheduler with 10K warm-up steps. We do not employ dropout or weight decay. All models are trained

using bfloat16 precision. Given that even models achieving 100% in-distribution accuracy can benefit from further training, the best checkpoint is selected based on performance at $8\times$ the in-distribution sequence length. We evaluate using exact match accuracy on 1,000 samples for each sequence length. All models are trained from scratch with 3 different random seeds, and we report the results from the best-performing run.

Table 3: Synthetic task details and hyperparameters.

| Task | Samples | Batch | Vocab. | Layers | Hid. dim. |
|------|---------|-------|--------|--------|-----------|
| MQRAR | 20M | 128 | 256 | 2 | 256 |
| Copy | 20M | 128 | 32 | 2 | 256 |

## D.3 REAL-WORLD PRETRAINING DETAILS

**Model Configurations.** Our pretrained models are based on a standard decoder-only Transformer architecture. The specific hyperparameters for the 1B and 3B parameter models are detailed in Table 4. All models were trained using the same vocabulary and tokenizer for fair comparison.

Table 4: Model configurations for our 1B and 3B parameter models.

| Hyperparameter | 1B Model | 3B Model |
|----------------|----------|----------|
| Hidden Dimension ($d_{\mathrm{model}}$) | 2048 | 3072 |
| Number of Layers ($L$) | 24 | 32 |
| Number of Attention Heads | 32 | 32 |
| FFN Intermediate Size | 8192 | 8192 |
| Vocabulary Size | 32,000 | 32,000 |
| Activation Function | GeLU | GeLU |

**Evaluation Benchmark Details.** All real-world data evaluations were conducted using the LM Evaluation Harness framework (Gao et al., 2023). Below are brief descriptions of the benchmarks used in our pretraining evaluation (Table 2).

- ARC (AI2 Reasoning Challenge) (Clark et al., 2018): A collection of grade-school level, multiple-choice science questions. We report on both the Challenge (ARC-c) and Easy (ARC-e) sets.

- Hellaswag (Zellers et al., 2019): A commonsense reasoning benchmark that involves choosing the most plausible continuation of a given sentence.

- OBQA (OpenBookQA) (Mihaylov et al., 2018): An open-book question answering dataset that requires reasoning over a small set of common knowledge facts.

- PIQA (Physical Interaction QA) (Bisk et al., 2020): A commonsense reasoning benchmark focused on understanding physical interactions and choosing the more plausible of two given solutions.

- RACE (Lai et al., 2017): A large-scale reading comprehension dataset collected from English examinations for middle and high school students in China.

- Winogrande (Sakaguchi et al., 2021): An adversarial version of the Winograd Schema Challenge, designed to be robust against dataset biases for commonsense reasoning.

- Wikitext PPL (Perplexity): We measure the perplexity on the Wikitext-103 dataset (Merity et al., 2016), a standard benchmark for evaluating the language modeling capabilities of a model.

# USE OF LLM

We only apply LLM for checking spelling and grammar.

