# OpenReview forum: "Sieve Attention: Fusing Context-Aware Filtering and Sequential Allocation For Long Sequence"
_ICLR.cc/2026/Conference — ICLR 2026 Conference Withdrawn Submission_

### Official Review · Reviewer_FTWn · 2025-10-19

**Soundness:** 2
**Presentation:** 3
**Contribution:** 2
**Rating:** 2
**Confidence:** 4

**Summary:**

The paper introduces Sieve Attention, a new Transformer attention mechanism for handling long sequences. It combines content-based filtering using α-entmax with a sequential, recency-biased allocation step, removing the need for positional encodings. This design lets models focus on relevant information while preserving order sensitivity. Experiments show Sieve Attention outperforms existing methods in long-context modeling.

**Strengths:**

The paper is clearly written.

**Weaknesses:**

1. Limited Novelty: The idea to "first select, then attend" is not new for sparse attention. The authors can revisit the classic Reformer [1] paper, which first select the most relevant contents with hash projection, and then do recency-biased attention with sliding window. What's more, the time complexity of Reformer is strictly sub-quadratic with respect to the sequence length, which is better than the authors' sieve attention for long context;

2. Safety Concern: The motivation to have recency bias in attention does not align with some of the most important LLM capabilities, e.g. instruction following. For example, instructions on safety should never be forgotten throughout the generation process. However, if we apply the authors' sieve attention, owing to its recency-biased nature, it might be vulnerable to attack strategies like injecting misleading information in the middle of the context.

3. Limited Evaluation: The current evaluation largely focuses on synthetic tasks. The authors are expected to evaluate and report their results across more settings, e.g.

    - Full benchmark results of Ruler across 13 subtasks;
    - At least one real-world long-context benchmark (e.g. LongBench [2]), since the context length in Table 2 tasks are not very long.

[1]. Reformer: The Efficient Transformer. ICLR 2020.
[2]. LongBench: A Bilingual, Multitask Benchmark for Long Context Understanding. ACL 2024.

**Questions:**

N/A

---

### Official Review · Reviewer_FXWg · 2025-10-26

**Soundness:** 3
**Presentation:** 3
**Contribution:** 2
**Rating:** 6
**Confidence:** 3

**Summary:**

This paper investigates a sparse attention mechanism that combines α-antmax and stick-breaking attention. The main idea is to use α-antmax to compute attention scores, then select the top-k tokens to achieve sparsity, and finally apply stick-breaking attention to these tokens. Experimental results show that this method explicitly outperforms traditional softmax and stick-breaking attention in terms of extrapolation capability.

**Strengths:**

1. This paper explores an attention mechanism based on the combination of sparse α-antmax and stick-breaking, and conducts thorough experiments. Compared to the baseline, it achieves improvements in both in-domain and extrapolation settings.

**Weaknesses:**

1. Compared to stick-breaking attention, the improvement is relatively marginal: From Figure 5, I observe that the extrapolation curve is slightly better overall than that of stick-breaking attention, but the enhancement does not appear to be particularly significant.
2. In terms of methodology, the approach lacks novelty and is essentially a combination of existing methods: The overall idea is based on sparse α-antmax and stick-breaking attention, which lacks deeper insights.
3. Missing references: Some relevant length generalizable attention works are missing, such as https://arxiv.org/abs/2305.16300, https://arxiv.org/abs/2410.01651.

**Questions:**

see weaknesses

---

### Official Review · Reviewer_V5V7 · 2025-10-30

**Soundness:** 2
**Presentation:** 2
**Contribution:** 2
**Rating:** 2
**Confidence:** 4

**Summary:**

This paper introduces Sieve Attention, a two-stage attention mechanism designed to address the limitations of standard softmax-based attention in long-context generalization. The method combines a sparse content-based filtering stage (using α-entmax) with a sequential allocation stage (using a stick-breaking process) to improve focus and reduce reliance on positional encodings. The authors provide theoretical analysis and experimental results on synthetic and real-world benchmarks to demonstrate the effectiveness of their approach.

**Strengths:**

- The two-stage design of Sieve Attention is intuitively appealing and aligns with the goal of decoupling content-based relevance from sequential prioritization.
- Extensive experiments on both synthetic and real-world benchmarks (e.g., MQRAR, RULER) are conducted, showing consistent improvements over strong baselines.
- The hardware-efficient implementation and ablation studies add practical value and credibility to the proposed method.

**Weaknesses:**

1. The core components of Sieve Attention—α-entmax and the stick-breaking process—are not novel. Both have been previously introduced and studied in the literature [1,2]. The combination of these two existing techniques does not constitute a significant conceptual contribution.
2. The claim that the entropy of the final attention distribution is strictly lower than that of the candidate distribution relies on the assumption that the sigmoid function σ becomes saturated (i.e., σ(z) → 1). In practice, this saturation is not guaranteed, and the entropy reduction may not hold universally. This limits the generality of the theoretical claim.
3. While the authors propose a hardware-efficient implementation, Sieve Attention still underperforms FlashAttention in terms of throughput. Since FlashAttention is an optimized implementation of standard softmax attention without altering its mathematical form, the lower throughput of Sieve Attention suggests higher inherent computational complexity, which may limit its practicality for large-scale deployment.
4. The related works should include discussion and comparison with the broader landscape of sparse attention mechanisms designed for long-context modeling (e.g., [3-9]).

**Questions:**

NA

---

### Official Review · Reviewer_PXeG · 2025-10-31

**Soundness:** 2
**Presentation:** 3
**Contribution:** 2
**Rating:** 2
**Confidence:** 5

**Summary:**

This paper introduces Sieve Attention, a attention mechanism designed to address the long-context generalization limitations of the standard Transformer. The proposed method decouples the attention process into two stages: (1) a content-based filtering stage using α-entmax to select a sparse set of candidate tokens, and (2) a sequential allocation stage using a stick-breaking process on this candidate set to assign final weights with a recency bias. The authors argue that this design eliminates the need for external positional encodings and provides superior length extrapolation. The paper includes theoretical analysis and empirical evaluations on synthetic tasks, language model pretraining, and long-context benchmarks to support these claims.

**Strengths:**

- The paper clearly identifies and motivates the core challenges in long-context modeling: attention dispersion from dense softmax and the brittleness of positional encodings.
- The two-stage design of Sieve Attention is explained clearly and illustrated effectively with diagrams.
- The discussion on a hardware-efficient implementation, including online α-entmax and in-SRAM sequential allocation, demonstrates a consideration for practical deployment.

**Weaknesses:**

- The core technical components of Sieve Attention are not novel. Both the α-entmax sparse activation function [1] and the stick-breaking process for attention [2] are existing methods. The primary contribution of this work is their specific combination. While this combination is non-trivial, it does not constitute a significant conceptual leap.
- The claim that the entropy of the final distribution \(H(a_j)\) is strictly less than that of the candidate distribution \(H(c_j)\) is not fully substantiated. The analysis relies on the scenario where \(\sigma(z_{i_l,j}) \to 1\) for a recent token, causing weights for earlier candidates to vanish. However, it does not account for the saturation properties of the sigmoid function. In practice, if multiple candidates have high, saturated logits, the resulting distribution \(a_j\) may not be significantly more concentrated than \(c_j\), and the entropy reduction may be minimal. The analysis would benefit from a more rigorous treatment of this dynamic.
- The condition for filtering out a distractor token—\((\alpha-1)z_d \leq \tau(z_j)\)—is presented in an idealized, noise-free setting. The proposition assumes that a single high-scoring relevant token \(t_f\) can raise the global threshold \(\tau(z_j)\) sufficiently to filter all distractors. In realistic scenarios with complex, multi-modal score distributions and numerous distractors, this global threshold may not be as effective, and the theoretical guarantee may not hold. An analysis incorporating noisy score distributions would strengthen this claim.
- The paper acknowledges that Sieve Attention's throughput is lower than the highly optimized FlashAttention. Given that FlashAttention is an implementation of standard Softmax attention, this performance gap suggests that the two-stage Sieve Attention algorithm has a higher inherent computational and/or memory complexity. While the complexity is argued to scale with the candidate set size \(s_j\), the overhead of the filtering stage and the sequential allocation on non-contiguous data appears non-negligible in practice. This raises concerns about its efficiency as a drop-in replacement for large-scale training and inference.

---

[1] Sparse sequence-to-sequence models. ACL 2019
[2] Scaling Stick-Breaking Attention: An Efficient Implementation and In-depth Study. 2024

**Questions:**

NA

---

### Note · Authors · 2025-11-12

I have read and agree with the venue's withdrawal policy on behalf of myself and my co-authors.